# Novel Carboxylation Method for Polyetheretherketone (PEEK) Surface Modification Using Friedel–Crafts Acylation

**DOI:** 10.3390/ijms242115651

**Published:** 2023-10-27

**Authors:** Xinghui Lyu, Ryuhei Kanda, Susumu Tsuda, Yoshiya Hashimoto, Takamasa Fujii, Kosuke Kashiwagi

**Affiliations:** 1Department of Fixed Prosthodontics and Occlusion, Osaka Dental University, 8-1 Kuzuhahanazonocho, Hirakata 573-1121, Osaka, Japan; lyu-x@cc.osaka-dent.ac.jp (X.L.); taka-f@cc.osaka-dent.ac.jp (T.F.); fpd-kk@cc.osaka-dent.ac.jp (K.K.); 2Division of Creative and Integrated Medicine, Advanced Medicine Research Center, Translational Research Institute for Medical Innovation (TRIMI), Osaka Dental University, 8-1 Kuzuhahanazonocho, Hirakata 573-1121, Osaka, Japan; yoshiya@cc.osaka-dent.ac.jp; 3Department of Chemistry, Osaka Dental University, 8-1 Kuzuhahanazonocho, Hirakata 573-1121, Osaka, Japan; tsuda-s@cc.osaka-dent.ac.jp; 4Department of Biomaterial, Osaka Dental University, 8-1 Kuzuhahanazonocho, Hirakata 573-1121, Osaka, Japan

**Keywords:** polyetheretherketone, friedel–crafts acylation, carboxyl group, chemical modification, pentadecafluorooctylamide

## Abstract

Recently, polyetheretherketone (PEEK) has shown promising dental applications. Surface treatment is essential for dental applications owing to its poor surface energy and wettability; however, no consensus on an effective treatment method has been achieved. In this study, we attempted to carboxylate PEEK sample surfaces via Friedel–Crafts acylation using succinic anhydride and AlBr_3_. The possibility of further chemical modifications using carboxyl groups was examined. The samples were subjected to dehydration–condensation reactions with 1*H*,1*H*-pentadecafluorooctylamine and *N*,*N*’-dicyclohexylcarbodiimide. Furthermore, the sample’s surface properties at each reaction stage were evaluated. An absorption band in the 3300–3500 cm^−1^ wavenumber region was observed. Additionally, peak suggestive of COOH was observed in the sample spectra. Secondary modification diminished the absorption band in 3300–3500 cm^−1^ and a clear F1s signal was observed. Thus, Friedel–Crafts acylation with succinic anhydride produced carboxyl groups on the PEEK sample surfaces. Further chemical modification of the carboxyl groups by dehydration-condensation reactions is also possible. Thus, a series of reactions can be employed to impart desired chemical structures to PEEK surfaces.

## 1. Introduction

In recent years, the trend of metal-free dentistry has accelerated against the backdrop of soaring metal prices, increasing esthetic awareness, and concerns related to metal allergies. Under these circumstances, polyetheretherketone (PEEK) has attracted attention as a new candidate for dental material [1,2,3,4]. PEEK is a thermoplastic polymer with excellent mechanical strength, chemical, heat, and wear resistance, and biosafety [5,6]. In dentistry, PEEK is used in abutments or temporary restorations for dental implants [7,8,9], frameworks for removable dentures [10,11,12], retentive devices such as clasps [13,14] or double crowns [15,16], as well as frameworks for fixed prosthesis [17,18,19] and post and cores [20,21] as restorative or prosthetic materials.

Nevertheless, PEEK exhibits negative characteristics which lead to adhesion or bioactivity related issues, including low surface energy and poor wettability [6,22]. Therefore, using PEEK in adhesive dental procedures is restricted to restorative or prosthetic applications, in particular during oral and maxillofacial surgery. Considering this, a wide variety of approaches have been investigated to better elucidate the material’s possible dental applications [22]. Recently, chemical modification methods have been adopted in industrial fields to improve the adhesion of PEEK. Zhang et al. proposed a method to introduce epoxy groups of a polymer brush by generating free radicals in PEEK under ultraviolet irradiation [23]. Furthermore, Miyagaki et al. introduced epoxy groups to a PEEK surface using a chemical reaction system based on the Friedel–Crafts reaction and reported improved bond strength with epoxy resin adhesives [24]. Among these, the theoretical method using the Friedel–Crafts acylation reaction has the potential to provide any chemical structure to the aromatic ring region in PEEK, making it possible to develop modified PEEK with chemically customized surface properties for each application in the dental field. However, the mechanisms by which various treatments affect the surface properties of PEEK remain unclear. Furthermore, an established and optimized PEEK surface treatment method which can be used in various dental applications is lacking. Therefore, the establishment and development of treatment methods based on the chemical modification of PEEK surfaces could greatly expand the applications of PEEK in dentistry and provide much needed treatment options.

In this study, Friedel–Crafts acylation [25] using succinic anhydride was attempted to introduce carboxyl groups as initial functional groups for modifying PEEK surfaces with any chemical structure. The effects of the treatment on surface properties were evaluated using X-ray photoelectron spectroscopy (XPS), Fourier-transform infrared spectroscopy (FT-IR), and water contact angle measurements. Furthermore, 1*H*,1*H*-pentadecafluorooctylamine and *N*,*N*’-dicyclohexylcarbodiimide (DCC) were used in dehydration–condensation reactions to investigate the possibility of further surface modification. To the best of our knowledge, no other research has attempted the Friedel–Crafts acylation of PEEK using succinic anhydride. This innovative chemical treatment method introduced biocompatible carboxyl groups onto the PEEK surface in a single step. If further chemical modifications are possible by the dehydration–condensation reaction of the carboxyl groups, the possibility of imparting arbitrary chemical structures to the PEEK sample surface is expected.

## 2. Results

### 2.1. Materials, Reagents, and Experimental Protocol

The materials and reagents, chemical treatment schemes, and experimental protocol used and followed for the experiment are presented in Table 1 and Figure 1A,B, respectively. Three of the ten samples, pretreated with 98% concentrated sulfuric acid, were randomly selected and sonicated with chloroform (CHCl_3_), dilute hydrochloric acid, and distilled water for 5 min each, then dried under vacuum for 1 d at 30 °C. These samples were used as control groups for various surface analyses. The other six samples were subjected to a first-step reaction and designated as the Friedel–Crafts acylation group (FC); surface analysis was performed after washing and drying the samples, similar to the control group. After completion of the analysis, six samples from the FC group were subjected to a second-step reaction of dehydration–condensation and were designated as the secondary modified group (M). After washing each sample of group M with CHCl_3_, acetone, and distilled water, they were dried under vacuum at 30 °C for 1 d. Thereafter, surface analysis was performed. The remaining sample, pretreated with 98% concentrated sulfuric acid, only underwent the second-step reaction. It was used as the negative control (N.C.) and was evaluated through XPS and water contact-angle measurements.

### 2.2. Carboxylation of PEEK Surface by Friedel–Crafts Acylation with Succinic Anhydride

Fourier transform infrared (FT-IR) analysis was performed to investigate the effect of Friedel–Crafts acylation with succinic anhydride on the chemical bonding state of the PEEK surface. FT-IR spectra of the control and FC samples are shown in Figure 2A. In both samples, absorption bands were observed at approximately 1700, 1600, 1500, and 1200 cm^−1^ due to carbonyl stretching and contraction, benzene ring stretching and contraction or in-plane vibration, and aromatic ether, respectively. Conversely, almost all samples in the FC group showed a mild absorption band at approximately 3300–3500 cm^−1^, which corresponded to the stretching motion of OH derived from the carboxyl groups. Thus, carboxyl groups were introduced onto the PEEK surface by Friedel–Crafts acylation.

Thereafter, the effects of the treatment on the chemical shifts of C1s and O1s orbitals were evaluated using narrow-scan XPS analysis. Figure 2B shows a representative superimposed image of a spectra for each sample from the control and FC groups. All FC samples showed a tendency to move toward a decreasing intensity peak at approximately 291 eV in the C1s relative to the main peak near 284 eV, compared with the control group. Furthermore, the shape of the peaks changed from bimodal to unimodal in the O1s.

In addition, waveform separation into various carbon species was performed by curve fitting the C1s spectra of the control and FC groups. A representative example of curve fitting (Figure 2C), mean value (% area), and standard deviation (SD) of the area ratio of each peak are listed in Table 2. In the FC group, the peak at approximately 287.9 eV, indicative of COOH, appeared with an average % area of 2.64. Conversely, the % area of the peak at approximately 290.4 eV relatively decreased from 6.76 to 3.81 due to π-bonding. These results suggested the presence of COOH on the FC sample surface, confirming that the PEEK surface was carboxylated by Friedel–Crafts acylation with succinic anhydride.

### 2.3. Confirmation of Secondary Modification of Carboxylated PEEK

For secondary modification of the carboxylated PEEK samples, a dehydration–condensation reaction between the FC samples and 1*H*,1*H*-pentadecafluorooctylamine was performed to obtain the modified PEEK samples (M). Wide-scan XPS analysis was performed on each of the six samples in the M group. Representative spectra of the M group and N.C. are shown in Figure 3A. Compared to N.C., a prominent F1s signal was observed at approximately 685 eV in the M group spectrum. In addition, an F-Kll signal was observed at approximately 832 eV. The spectra of all M-group samples are shown in Appendix A. A clear F1s signal was observed in all samples of the M group. The % area of each spectrum was quantified using the analysis software (PHI MultiPak Ver. 9.4.0.7, Ulvac-PHI, Kanagawa, Japan). The percentage of F1s in the M group was significantly larger than that in the N.C. (Figure 3B). Thus, alkyl fluoride groups were introduced on the carboxylated PEEK surface by the dehydration–condensation reaction, indicating the possibility of further chemical modification of the carboxylated PEEK surface.

Figure 3C shows the FT-IR results for each reaction step; the absorption band at approximately 3300–3500 cm^−1^, indicating the carboxyl-group-derived OH stretching observed in the FC group, disappeared in the M group. Furthermore, the absorption bands in the wavenumber region at approximately 3300–3500 cm^−1^ disappeared or weakened in all experimental samples (Appendix A). Thus, the dehydration–condensation reaction of aliphatic amines with the carboxyl groups on the treated PEEK surface, which were imparted through Friedel–Crafts acylation, formed an amide bond, such that the OH at the COOH terminus was no longer detectable. Therefore, secondary chemical modifications were initiated by the carboxyl group imparted by Friedel–Crafts acylation with succinic acid.

### 2.4. Water Contact Angle Measurement at Each Reaction Step

The qualitative findings of droplets (Figure 4) and the contact angle measurements for each reaction phase are listed in Table 3. The contact angles of the control, FC, and M groups were 84.54 ± 2.98°, 84.5 ± 4.24°, and 90.61 ± 5.4°, respectively. The increase in contact angle of the M group was observed to be more than that of the other groups. Thus, the secondary modification imparted long-chain alkyl fluoride groups to the PEEK sample surface, causing a change in physicochemical properties of the PEEK surface and a decrease in wettability.

### 2.5. X-ray Diffraction and Electron Microscopy Findings before and after Chemical Treatments

X-ray diffraction (XRD) analysis was performed to determine the effects of the chemical treatments on the crystal structure of the PEEK samples. Both the control and the M groups exhibited diffraction peaks at 2θ = 18.9°, 20.8°, 22.8°, and 28.8° (Figure 5A), which represented the [110], [111], [200], and [211] lattice planes of the original PEEK, respectively [24,26]. Thus, no change in the crystal structure of the samples was observed before and after the reaction.

In addition, scanning electron microscopy (SEM) of the surfaces of the original PEEK (without the 98% concentrated sulfuric acid pretreatment), control, and M samples was performed (magnified 5000×) to investigate differences in the morphological characteristics of the surfaces before and after the reaction. Porous structures were observed in the samples of both the control and M groups after treatment with 98% concentrated sulfuric acid. No significant differences in the surface morphological characteristics of the sample in the control and M groups were observed (Figure 5B). Thus, no significant change in the morphology of the PEEK surfaces was caused by the Friedel–Crafts acylation and condensation reactions.

## 3. Discussion

Since PEEK is a relatively new material in dentistry, knowledge of surface treatment methods and adhesive techniques is limited. Acid etching [27,28,29] and sandblasting [30,31,32], which are common physicochemical treatments for PEEK, have been reported to effectively enhance bond strength and positively affect the increase in bond area owing to partial erosion by acid or the formation of irregularities and mechanical mating force by abrasive grain impact, respectively. Influenced by the improvement of wettability of the adherend surface, removal of organic and other contaminants, and changes in chemical properties due to the addition of functional groups, few studies have considered plasma treatment as an effective approach [33,34,35,36]. Conversely, nucleophilic addition reactions to carbonyl groups in PEEK molecules [37] and modification of aromatic ring structures by the Friedel–Crafts reaction [24] have been reported as methods for the precise chemical modification of PEEK. For the first time, a method for the carboxylation of PEEK surfaces by Friedel–Crafts acylation with succinic anhydride for aromatic ring structures has been proposed in this paper.

First, the PEEK samples were treated with 98% concentrated sulfuric acid for 3 min to increase the surface area of the reaction layers. Thereafter, FT-IR, XPS, and XRD analyses were performed before and after the etching treatment. The results showed no changes in the chemical characteristics of the PEEK sample surfaces (data not shown). Therefore, concentrated sulfuric acid etching treatment at room temperature for a short time, which is a relatively frequent treatment in dentistry [2,27,28,29], changed the surface morphology of the PEEK samples to a porous structure but had little effect on the chemical surface properties.

The FT-IR results showed that the Friedel–Crafts acylated samples exhibited absorption bands at an approximately 3300–3500 cm^−1^ wavenumber range compared to the samples treated with only sulfuric acid. Thus, the absorption was due to the OH-stretching motion [37] of the succinic anhydride-derived carboxyl group (COOH) on the sample surface. However, since the COOH-derived C=O peak overlapped with the peak of the existing carbonyl group in the PEEK molecule, the peak was not distinct in the FT-IR spectra. The spectra were lost or weakened by the dehydration–condensation reaction with 1*H*,1*H*-pentadecafluorooctylamine, as was reflected by the formation of an amide bond between the COOH terminus on the PEEK sample side and the amino group on the aliphatic amine side, resulting in an undetectable OH group at the carboxyl terminus. Additionally, waveform separation of the C1s spectrum on the PEEK sample surface after Friedel–Crafts acylation, observed during XPS analysis, directly suggested the presence of a carboxyl group. Previous reports examining the introduction of carboxyl groups on PEEK surfaces showed a spectral chemical shift pattern [34,35], similar to our study. In addition, the F1s signal derived from the reaction substrate was clearly detected on the surface of the samples after secondary modification. Contrarily, no such signals were detected in the case of N.C., which underwent a secondary modification reaction without Friedel–Crafts acylation. Thus, the series of reactions originating from the carboxyl groups were imparted to the sample surfaces by Friedel–Crafts acylation using succinic anhydride. Miyagaki et al. used 1*H*,1*H*-pentadecafluorooctylamine within a similar modification process as a detection system for Friedel–Crafts reaction [24]; thus, the second-step reaction of dehydration–condensation in our experiments is expected to be significant as a detection system for carboxyl groups. Furthermore, wettability evaluation using water contact angle measurements showed that the sample surfaces after the secondary modification were more hydrophobic than those at other reaction stages. This reflected the addition of long-chain alkyl fluoride groups derived from the reaction substrate to the sample surfaces via secondary modification, indicating that a series of treatments may change the physicochemical properties of the sample surface. In addition, the XRD and SEM images showed no clear differences between the control and M group samples used in the experiment, indicating no change in the crystallinity and morphological characteristics of the samples during the series of reactions. Thus, the surface corrosion resistance or mechanical properties of the samples were not affected.

The proposed chemical treatment method did not require special equipment and was inexpensive and simple. Furthermore, the reaction substrate, succinic anhydride, could be hydrated environmentally to produce succinic acid, which is a naturally occurring substance with excellent biosafety [38]. Since carboxylated surfaces were obtained in a single operation for the experiment, mass production through plantations can be considered in the future. Moreover, the development of a variety of modified PEEKs to improve adhesion or biocompatibility is possible because Friedel–Crafts acylation using succinic anhydride as the first and second steps of the condensation reaction can give an arbitrary chemical structure.

## 4. Materials and Methods

### 4.1. Preparation of PEEK Samples

PEEK samples of 10 mm per side and 5 mm in thickness were prepared and polished under running water to #800 using a water-resistant polishing paper. Each sample was ultrasonically cleaned in purified water for 5 min and subsequently dried. To increase the surface area of the treated surface, the samples were etched with 98% concentrated sulfuric acid for 3 min, rinsed, dried under vacuum, and used in subsequent experiments.

### 4.2. Friedel–Crafts Acylation (First-Step Reaction)

Succinic anhydride (17 mg, 0.17 mmol) was dissolved in 5 mL of superhydrated CHCl_3_ under a nitrogen atmosphere, and the PEEK samples were immersed. In addition, 0.3 mL of aluminum bromide solution (1.0 M AlBr_3_ in dibromomethane) as Lewis acid was added, and the reaction was heated to 55 °C for 24 h. The collected samples were sonicated with CHCl_3_, dilute hydrochloric acid, and distilled water for 5 min each, and dried at 30 °C under vacuum for 24 h.

### 4.3. Dehydration–Condensation Reaction (Second-Step Reaction)

1*H*,1*H*-pentadecafluorooctylamine (79.8 mg, 0.2 mmol) and DCC (41.3 mg, 0.2 mmol) were dissolved in 2 mL of CHCl_3_ under a nitrogen atmosphere. PEEK samples with or without (N.C.) for the first-step reaction were immersed in 2 mL of CHCl_3_, heated to 55 °C, and allowed to react with the solution for 24 h. The collected samples were washed twice with CHCl_3_, acetone, and distilled water by sonication for 5 min each, and dried at 30 °C under vacuum for 24 h.

### 4.4. Water Contact Angle Measurement

The contact angle of each sample with water was measured using a contact-angle meter (LSE-ME2; Nick, Saitama, Japan). After confirming the stability of the droplet shape, the droplet was photographed horizontally with an attached digital camera, and the contact angle was measured using contact-angle measurement software (i2win.n Ver. 3.4.0.2, Nick). The same procedure was performed three times for each sample, and the average value was used as the contact angle with water for each sample.

### 4.5. Fourier Transform Infrared Spectroscopy (FT-IR)

FT-IR spectrophotometer (IRAffinity-1S, Shimadzu, Kyoto, Japan) and accompanying analysis software (LabSolutions IR Ver. 2.20, Shimadzu) were used as an attenuated total reflection method (ATR) to investigate the effect of each surface treatment on the chemical bonding state of the PEEK sample surfaces. The analysis was performed twice for each arbitrary measurement point. Measurement conditions were set to 32 integrations, a wavenumber range of 400–4000 cm^−1^, and a resolution of 4 cm^−1^.

### 4.6. X-ray Photoelectron Spectroscopy (XPS) 

XPS analysis was performed to confirm the chemical state of the samples after various surface treatments by analyzing their elemental compositions, peak shapes, and positions. The analysis was performed using a PHIX-tool (Ulvac-Phi, Kanagawa, Japan) with Al Kα radiation at 15 kV and 4 W, an analysis range of 1 α m, and a take-off angle of 45°. The setup conditions for wide scan analysis were set to a pass energy of 280 eV, a step size of 0.25 eV, and 10 integrations. The setup conditions for the narrow-scan analysis were set for the C1s (278–298 eV) and O1s (523–543 eV) spectra with a pass energy of 69 eV, a step size of 0.125 eV, and 200 integrations. Three arbitrary points were selected as measurement sites for each sample. The obtained data were analyzed using analysis software (MultiPak Ver. 9.4.0.7, Ulvac-Phi), and the various data were quantified by curve fitting and area ratio calculations.

### 4.7. X-ray Diffraction Analysis (XRD)

XRD analysis was performed to determine the effects of various treatments on the crystal structure of the PEEK samples before and after treatment. A LabX XRD-6000 (Shimadzu) instrument was used for the analysis. The control and M groups were selected as target samples. Tube voltage and current were set to 40 kV and 30 mA, respectively, and analysis was performed using Cu Kα radiation at a scan speed of 2 deg/min.

### 4.8. Observation of Surface Morphology by Scanning Electron Microscope (SEM)

After osmium coating, the surface morphologies of the samples were observed using a SEM (SEM; S-4800, Hitachi, Tokyo, Japan).

### 4.9. Statistical Analysis

EZR software Ver. 1.61 was used for statistical analysis of the quantitative data. All quantitative data were expressed as means and SDs. *t*-tests were conducted for comparison of two groups, analysis of variance (ANOVA) with each chemical reaction stage as a factor for three or more groups, and Tukey’s multiple comparison test was used as a post test (α = 0.05).

## 5. Conclusions

The Friedel–Crafts acylation reaction on PEEK sample surfaces with succinic anhydride was found to impart carboxyl groups to the PEEK sample surfaces. Further, secondary modification was possible by dehydration–condensation reactions on the imparted carboxyl groups. These findings suggest the possibility of imparting arbitrary chemical structures to PEEK surfaces using a series of reactions. Therefore, based on the findings of this study, we conclude that PEEK modified with optimized surface properties can be obtained for various applications in the field of dentistry.

## Figures and Tables

**Figure 1 ijms-24-15651-f001:**
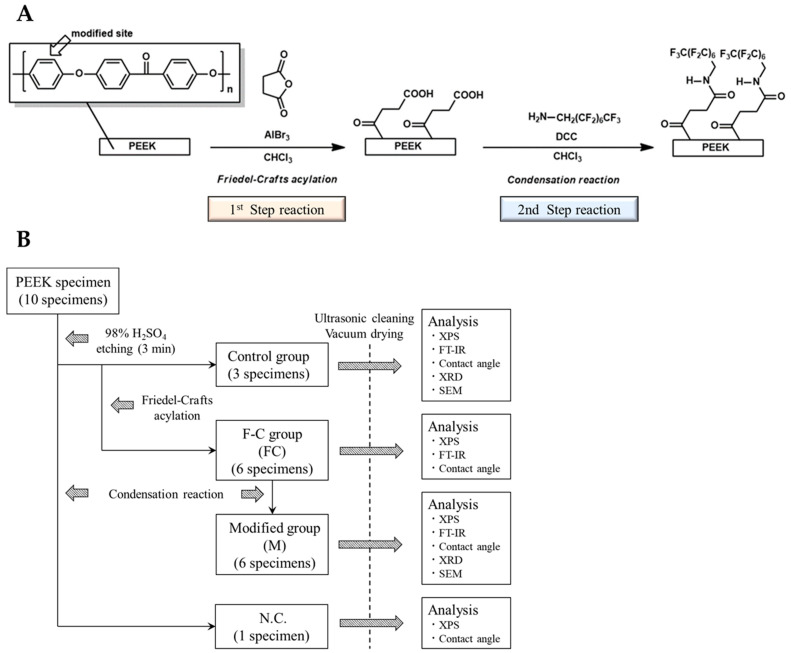
(**A**) Chemical treatment scheme for PEEK samples. (**B**) Experimental protocol.

**Figure 2 ijms-24-15651-f002:**
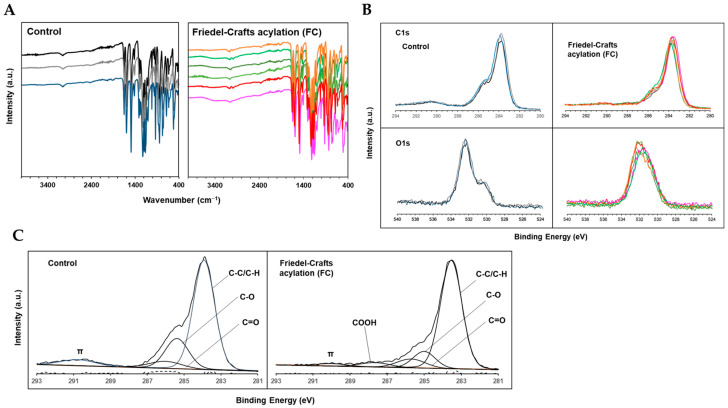
(**A**) FT-IR analysis results for the control and FC groups, with almost all samples in the FC group showing a broad absorption band in the 3300–3500 cm^−1^ region. (**B**) XPS results of C1s and O1s in the control and FC groups. C1s findings suggest a move toward a decreasing intensity peak at approximately 290.4 eV in all the FC group samples, while the O1s findings suggest change in the waveform from bimodal to unimodal. (**C**) Representative data of waveform separation by curve fitting for control and FC group, with the appearance of a peak suggestive of COOH at approximately 287.9 eV in the FC group samples.

**Figure 3 ijms-24-15651-f003:**
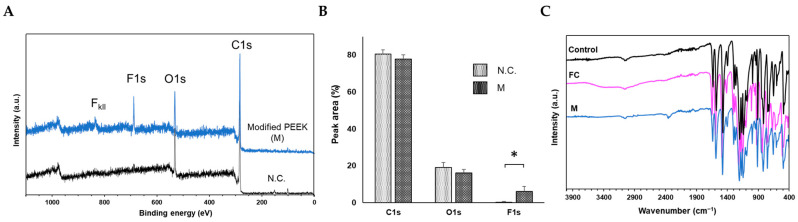
(**A**) Representative XPS data of N.C. (without Friedel–Crafts acylation) and M group (with Friedel–Crafts acylation) after dehydration–condensation reaction with 1*H*,1*H*-pentadecafluorooctylamine. A clear F1s signal is observed for the M group compared to that for the N.C. (**B**) % area ratio of each peak in N.C. and M group. Compared to N.C., the % of F1s markedly increases for the M group (* *p* < 0.05). (**C**) Representative FT-IR spectral data for each reaction step, with the M group showing a decrease in the absorption band at approximately 3300–3500 cm^−1^ observed in the FC group.

**Figure 4 ijms-24-15651-f004:**
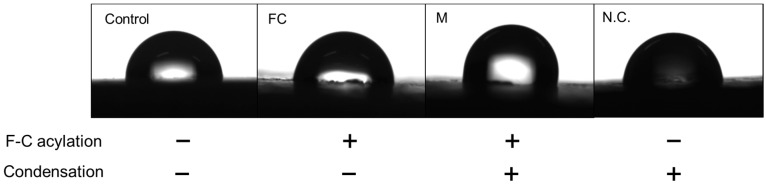
Wettability at each reaction step for different groups.

**Figure 5 ijms-24-15651-f005:**
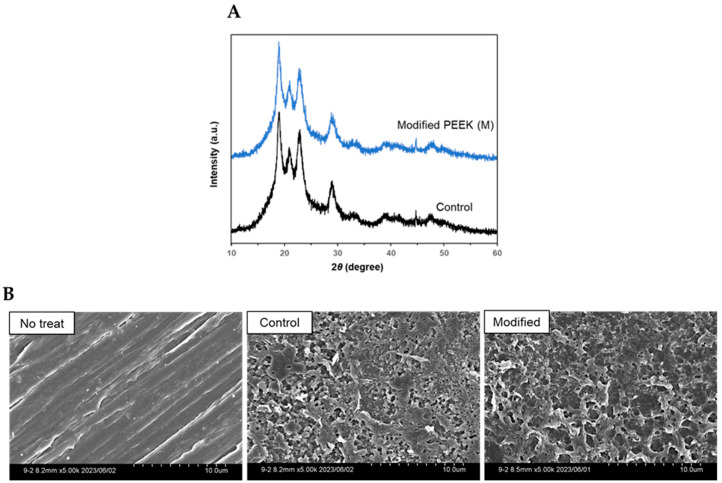
(**A**) XRD results of the samples in the control and M groups. No difference between the two samples is observed. (**B**) SEM images (×5000) of samples in the non-treated (Not treated), control (98% concentrated sulfuric acid treatment only), and M groups. Both control and M groups show porous surface morphology due to 98% concentrated sulfuric acid etching, but no significant differences in the morphologies of control and M groups are observed.

**Table 1 ijms-24-15651-t001:** Materials or reagents for chemical treatments.

Material or Reagent	Manufacture	CAS No.
PEEK	Victrex (Thornton Cleveleys, UK)	-
Succinic anhydride	TCI (Tokyo, Japan)	108-30-5
Aluminium bromide anhydorous (1.0 M in dibromomethane)	Sigma-Aldrich (St. Louis, MO, USA)	7727-15-3
1*H*,1*H*-Pentadecafluorooctylamine	Wako (Osaka, Japan)	307-29-9
*N*,*N*’-Dicyclohexylcarbodiimide	TCI	583-75-0

**Table 2 ijms-24-15651-t002:** Area ratios for each curve obtained from the analysis of the control and FC group samples. The number in () represents standard deviation; the ratio of COOH increased to 2.64 in the FC group (* *p* < 0.05), while the area ratio of π decreased relatively (^†^ *p* < 0.05).

Carbon Species	B.E (eV)	Mean Value of % Area	
		Control	Friedel–Crafts Acylation (FC)
C-C/C-H	283.7	67.52 (2.04)	72.57 (4.00)
C-O	285.2	19.09 (4.68)	12.10 (3.18)
C=O	285.9	6.62 (2.08)	8.88 (2.07)
COOH	287.9	0.00 *	2.64 (1.18) *
π	290.4	6.76 (1.06) ^†^	3.81 (1.54) ^†^

**Table 3 ijms-24-15651-t003:** Water contact angles at each reaction stage. Superscripts on the values of contact angles indicate differences between the values. M group shows a larger contact angle than the other groups (*p* < 0.05).

Sample Groups	Contact Angle (°)
Control	84.54 ± 2.89 ^a^
FC	84.50 ± 4.24 ^a^
M	90.61 ± 5.40 ^b^

## Data Availability

Not applicable.

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
