# Peer review of "Novel Carboxylation Method for Polyetheretherketone (PEEK) Surface Modification Using Friedel–Crafts Acylation"

_ijms, 2023, doi:10.3390/ijms242115651_

Round 1

Reviewer 1 Report

Comments and Suggestions for Authors

This work is well done and can be accetpted after minor remarks:

-Did authors use stock sample of chloroform or with additional purification (manufactured CHCl3 can be contains trace of carbon acids)?

-Authors wtirte 'Additionally, waveform separation of the C1s spectrum on the PEEK sample surface after Friedel–Crafts acylation, observed during XPS analysis, directly suggested the presence of a carboxyl group." Did authors sure that for C1s spectra the separation is observed? O1s is clearly change waveform after acylation but "flattening of peak" for C1s spectrum looks very vague. I think this moment must be cleared up.

Reviewer 2 Report

Comments and Suggestions for Authors

Authors presented valuable data on efficient modification of polyetheretherketone (PEEK), the obtained material is promising for application in dentistry. Friedel-Crafts acylation with succinic anhydride and subsequent formation of polyfluorinated amide groups are described in detail, samples are studied by XPS, XRD, FT-IR data. Some important additional measurements including water contact angle are provided. The research opens wide opportunities for the design of functionalized PEEK-based materials.

The manuscript can be published after minor revision. I would recommend including a phrase and a reference concerning the choice of polyfluorinated amine. In the keywords “carboxyl” would be more correct, and pentadecafluorooctylamides are worth mentioning as keyword.

Reviewer 3 Report

Comments and Suggestions for Authors

The article presents a procedure for improving a new candidate dental material, namely polyetheretherketone (PEEK) by modifying its surface.

            Most of the 40 references in the Introduction load the material unacceptably and do not bring useful elements to the study that follows.

            The structure of PEEK contains benzene rings that can be electrophilic substituted; one of these reactions is the Friedel-Crafts reaction. The authors relied on this reaction and acylated the benzene nucleus with succinic anhydride in the presence of AlBr3 in chloroform solution. Next, the carboxyl groups are subjected to dehydration–condensation reactions with 1H,1H-pentadecafluorooctylamine and N,N'-dicyclohexylcarbodiimide. The experimental procedures are well chosen and seem encouraging for obtaining the desired results in the practical application of the procedure. The results as well as the characterization of the produced materials were correctly evaluated using Fourier-transform infrared spectroscopy (FT-IR), X-ray photoelectron spectroscopy (XPS) and water contact angle measurements were compared with a negative control sample (N.C.) .

            The brief analysis made above allows to assume that PEEK modified with optimized surface properties can be used for various applications in the field of dentistry. As a conclusion the paper can be published with the modification of the Introduction.
